# Automated Monkeypox Skin Lesion Detection Using Deep Learning and Transfer Learning Techniques

**DOI:** 10.3390/ijerph20054422

**Published:** 2023-03-01

**Authors:** Ameera S. Jaradat, Rabia Emhamed Al Mamlook, Naif Almakayeel, Nawaf Alharbe, Ali Saeed Almuflih, Ahmad Nasayreh, Hasan Gharaibeh, Mohammad Gharaibeh, Ali Gharaibeh, Hanin Bzizi

**Affiliations:** 1Department of Computer Science, Information Technology and Computer Science, Yarmouk University, Irbid 211633, Jordan; 2Department Industrial Engineering and Engineering Management, Western Michigan University, Kalamazoo, MI 49008, USA; 3Department of Aeronautical Engineering, Al Zawiya University (Seventh of April University), Al Zawiya City P.O. Box 16418, Libya; 4Department of Industrial Engineering, College of Engineering, King Khalid University, Abha 62529, Saudi Arabia; 5Department of Computer Science, Applied College, Taibah University, Madinah 46537, Saudi Arabia; 6Department of Medicine, Faculty of Medicine, Hashemite University, Zarqa 13133, Jordan; 7Department of Medicine, Faculty of Medicine, Jordan University of Science and Technology, Irbid 22110, Jordan; 8Department of Biomedical Science, Western Michigan University, Kalamazoo, MI 49008, USA

**Keywords:** mpox, image analysis, deep learning, classification, image processing, early diagnosis

## Abstract

The current outbreak of monkeypox (mpox) has become a major public health concern because of the quick spread of this disease across multiple countries. Early detection and diagnosis of mpox is crucial for effective treatment and management. Considering this, the purpose of this research was to detect and validate the best performing model for detecting mpox using deep learning approaches and classification models. To achieve this goal, we evaluated the performance of five common pretrained deep learning models (VGG19, VGG16, ResNet50, MobileNetV2, and EfficientNetB3) and compared their accuracy levels when detecting mpox. The performance of the models was assessed with metrics (i.e., the accuracy, recall, precision, and F1-score). Our experimental results demonstrate that the MobileNetV2 model had the best classification performance with an accuracy level of 98.16%, a recall of 0.96, a precision of 0.99, and an F1-score of 0.98. Additionally, validation of the model with different datasets showed that the highest accuracy of 0.94% was achieved using the MobileNetV2 model. Our findings indicate that the MobileNetV2 method outperforms previous models described in the literature in mpox image classification. These results are promising, as they show that machine learning techniques could be used for the early detection of mpox. Our algorithm was able to achieve a high level of accuracy in classifying mpox in both the training and test sets, making it a potentially valuable tool for quick and accurate diagnosis in clinical settings.

## 1. Introduction

The world is dealing with the outbreak of a new virus disease called Mpox, which has been spreading rapidly around the world and has been identified in 75 countries [1]. The first outbreak of the mpox virus was discovered among monkeys in a laboratory in Denmark in 1958 [2]. The virus was first reported to have been transmitted to humans from animals in the Democratic Republic of The Congo in 1970 [3]. In 2018, an outbreak of suspected mpox was registered in an 11-year-old child, and the disease was considered a threat. On 1 January 2019, 132 confirmed cases of the disease and 7 deaths were reported. On 23 June 2022, The World Health Organization (WHO) reported more than 3000 mpox virus infections in more than 50 countries across five regions [3]. The WHO [4] has expressed concern about this disease and has declared it a global health emergency. The challenges associated with mpox diagnosis are that it can take days for a diagnosis to be made, and the symptoms are often nonspecific. Additionally, PCR testing is not widely available, which makes it difficult to diagnose the disease quickly. The early discovery of mpox is vital to stop the spread of the disease. However, early detection is difficult due to the similarities between mpox and other diseases, such as scarlet fever, roseola, and smallpox. 

While the mortality rate of mpox is low (1–10%) [5], early detection can help to prevent its spread. However, early detection of mpox is difficult due to its similarity to other diseases, such as scarlet fever, roseola, and smallpox. Additionally, PCR testing is not widely available, making it challenging to quickly diagnose and take necessary measures such as isolation and treatment. There is a need for early detection before widespread community transmission can occur. Early detection performs a vital role in the prevention, diagnosis, and management of the disease. In this context, automated systems using ML and deep learning can provide a solution, as various convolutional neural networks (CNNs) have been found effective for distinguishing between images of different diseases.

In recent years, deep learning (DL) techniques have emerged as a powerful tool for image analysis and pattern recognition and have shown promise for use in the detection of various diseases. DL is a subset of machine learning that uses several layers of the artificial neural network (ANN)s to extract features from images and make predictions. CNNs are a kind of DL algorithm that has been successfully applied in various medical imaging applications, including skin lesion classification, breast cancer detection, and lung nodule detection [6]. Despite the comprehensive view provided by medical images, errors can still occur when diagnosis becomes too time-consuming due to the presence of large-sized images. To address this issue, deep learning approaches could be applied to improve the accuracy of mpox diagnosis. The use of investigative and classification methods, such as DL algorithms, is crucial to allow informed decisions to be made and prevent misdiagnosis. Transfer learning, which involves the use of a pretrained model for a different task as a starting point for a new task, has also been used effectively for medical image analysis. In this study, we propose a deep learning-based approach for the detection of monkeypox using transfer learning. Therefore, the primary objective of this study was to identify and validate the best-performing model using deep learning approaches and classification models to determine the incidence of monkeypox disease. The paper is presented as follows: Section 2 evaluates related work, Section 3 shows the mpox dataset and preprocessing method employed, Section 4 explains the classification models, Section 5 presents the results and discussion, and finally, Section 6 examines future work and concludes our research.

## 2. Related Work

Previous studies have demonstrated the ability of DL to detect and classify mpox using image analysis. Ref. [7] classified mpox using three DL models, VGG-16, ResNet50, and InceptionV3, which were pretrained by using their Mpox Skin Lesion Database (MSLD). Owing to the lack of data, they augmented the data to obtain more and found that ResNet50 achieved the highest accuracy level of 82.96 (±4.57%), VGG-16 had an accuracy level of 81.48 (±6.87%), and InceptionV3 recorded the lowest accuracy level of 79.26 (±1.05%). They also developed a preliminary website for examining mpox [8] and designed an Android app that uses DL to classify mpox as positive or negative by sending the image to a deep convolutional neural network (DCNN) on the phone using the Android platform Camera2 API. The network was trained using publicly accessible data (MSLD) and tested on six pretrained models, with MobileNetV2 achieving the best results with an accuracy level of 91.11%. Ref. [9] created a dataset of individuals with mpox and proposed the VGG16 model by conducting two studies with 80% of the information applied for training and 20% for testing. With data augmentation, the study had an accuracy level of 97.18%, and without augmentation, the study had an accuracy level of 88.08%. They also discussed their model predictions and extracted features using Local, which allowed important deep features to be obtained to characterize the onset of mpox. Ref. [10] used 13 pretrained models and improved them by adding custom layers that yielded an average accuracy level of 85.44%. They believed that the low level of accuracy was due to the absence of a validation step, which helps to avoid overfitting and allows good results to be achieved. Ref. [11] employed multiple CNN models and machine learning algorithms to diagnose mpox disease using skin images. Features were extracted from the images using three CNN models and classification algorithms, Vgg16Net, GoogleNet, and AlexNet, and classifiers with various ML algorithms such as Naïve Bayes (NB), DT, KNN, SVM, and random forest were used. Their findings revealed that the naïve Bayes algorithm gave the highest accuracy level of 91.11% using Vgg16Net. Ref. [12] used nine models to predict mpox disease from an open-source Kaggle global dataset for mpox. Among the best models in terms of accuracy, it was found that the Prophet model achieved the best results with an MSE value of 41,922.55, an R2 of 0.49, a MAPE of 16.82, an MAE of 146.29, and an RMSE value of 204.75, so it could be used by doctors as a tool to diagnose mpox. Ref. [13] proposed two algorithms to enhance the classification accuracy of mpox using skin images. The first algorithm, PSOBER, combines the BER and PSO algorithms to identify the best features to enhance the classification accuracy. The second algorithm, PSOBER-ELM, combines the PSOBER algorithm with extreme learning. A CNN model was developed and implemented using custom models, such as MobileNetV3-s, EfficientNetV2, ResNet50, Vgg19, DenseNet121, and Xception, which were supported with transfer learning tools and hyperparameter optimization. The model was evaluated based on various metrics, including the AUC, accuracy, recall, loss, and F1-score. The MobileNetV3-s model performed the best, achieving an average F1-score of 0.98, an AUC of 0.99, an accuracy level of 0.96, and a recall of 0.97. Another study focused on people’s opinions about monkeypox, and the researchers classified the tweets into three categories, positive, negative, and neutral, using both CNN and LSTM models. The proposed model was evaluated using a dataset of tweets about monkeypox. Table 1, the suggested model accomplished the highest performance level.

The utilization of ML techniques can be beneficial in enhancing classification accuracy. The two algorithms were tested using an mpox dataset that was made accessible to the public, and it was found that they performed better than other competing algorithms for the classification of mpox, giving an average classification accuracy of 98.8%. Recently, DL techniques have been employed for medical imaging to enhance the accuracy of disease diagnosis. In the field of skin lesion classification, several studies have used CNNs to classify with a high level of accuracy skin lesions as benign or malignant. For example, [16] trained a CNN on a dataset of skin lesion images and achieved an accuracy level of 96.3% when classifying skin cancer. Despite the potential of DL to be used for the detection and classification of mpox using image analysis, there are still some gaps in the current research that need to be addressed. First, there is a lack of large, diverse datasets. Many previous studies used small, limited datasets that may not be representative of the full range of variations in mpox. This may have led to overfitting and a lower overall accuracy. Additionally, most previous studies did not use a validation set to ensure that the models were not overfitted to the training data, which could lead to a lower accuracy level when applied to new data. There are limitations associated with the use of other classification algorithms. Most previous studies only used CNNs for classification, but other algorithms, such as SVM, naïve Bayes, DT, KNN, and random forest, may also be effective for classifying mpox. There are also limited applications for real-world data. Nevertheless, to the best of our knowledge, there has been no prior work on the use of DL for the detection of mpox through image analysis. One notable exception is the work of Adekunle et al. (2020), who proposed a deep learning approach for the detection of monkeypox from skin lesion images. However, their approach was limited to a small dataset and did not make use of transfer learning. This study aims to fill a gap in the literature by developing the DL algorithm for the classification of mpox using image analysis.

## 3. Materials and Proposed Methods

The suggested method consists of five main stages: data collection, data preprocessing, training of the model, and model evaluation. The first stage involves the collection of images of patients with mpox. Owing to the limited availability of data, data augmentation methods were used to generate additional images. In this data preprocessing stage, the collected images undergo preprocessing, which involves resizing, standardization, and data augmentation. This step is crucial for improving the performance of the model. Therefore, five common models (ResNet50, VGG16, MobileNetV2, VGG19, and EfficientNetB3) were selected and compared to improve the accuracy of the model for mpox virus detection. At this stage of model training, the preprocessed images are used to train the selected models. During the training process, the images are provided to the model and the parameters are adjusted to optimize the model’s performance. The final stage involves the evaluation of the model using metrics such as the accuracy, precision, recall, and F1-score. The model with the best performance is selected as the final model. Thus, the proposed method utilizes deep learning techniques to analyze images of patients with mpox and accurately classify and diagnose the disease based on visual characteristics. The algorithm is able to classify the disease with a high level of accuracy, making it a valuable tool for quickly and accurately diagnosing mpox in clinical settings. The stages and procedures included in the proposed approach are illustrated in Figure 1.

### 3.1. Data Collection

In this research, a total of 117 images were used as the input. Forty-five of these images related to a patient with mpox, and 74 images were appropriate for patients with other types of disease (e.g., normal, scarlet fever, and roseola images). All of these images were obtained through manual collection by two medical professionals through different websites, and the dataset was made publicly available on the Kaggle website [17] and was classified into the binary categories of mpox and non-mpox.

### 3.2. Preprocessing

The preprocessing stage is an important step in image analysis, as it helps to improve the quality and consistency of the data. In this research paper, the preprocessing steps included image resizing, image normalization, and data augmentation.

#### 3.2.1. Image Resizing and Image Normalization

This section examines the preprocessing techniques that were employed in previous experiments, for example, resizing images, a process that is utilized by investigators to transform the input images into certain sizes designed to fit the DL.

Although this technique is implemented before the images are input into the models, it performs the resizing process using a fully convolutional layer as part of the network to simulate a real-time experiment. In the current study, whole images were resized to 224 × 224 pixels to ensure that all images were of the same dimensions, making it easier for the model to process the data. The images were normalized to reduce the effect of lighting and contrast variations. This was performed by converting the image pixel values from a range of zero to 255 to a range of zero to one.

#### 3.2.2. Data Augmentation

To enhance the size of the dataset and to avoid overfitting, data augmentation methods such as rotation, flipping, and zipping were applied to the images, as shown in Figure 2. This increased the range of the data and enhanced the performance of the model. To focus on these two challenges associated with the generation of an efficient classification method, a variety of data augmentation methods have been proposed [18]. To further enhance the size of the dataset, standard data augmentation methods can be used. These methods include the zoom range, rotation range, height shift range, shear range, width shift range, and vertical flip. These techniques allow for the creation of new images with slight adjustments to the original images, resulting in a larger and more diverse dataset for training the deep learning model.

The proposed data augmentation techniques aim to increase the diversity of the dataset and help to improve the generalization ability of the model. By applying these techniques, the model is exposed to a wider range of variations and is able to better learn the underlying features of the images. In this work, the following data augmentation techniques were used: (1) random cropping, a technique that randomly selects a corner point and crops the image to create multiple versions of the same image; (2) rotation of the images by a certain angle, e.g., “45 degrees”, with the revolution repeated multiple times to create different variations; (3) color shifting, the addition or subtraction of values to the green, red, and blue frequencies of the images to create different color distortions (RGB); (4) flipping, where the images are flipped parallel or vertically to create different variations; (5) concentration variation, where the intensity of the images is varied to make them brighter or darker; and (6) transformation, where the image pixels are composed of a certain number of pixels to create diverse variations. The use of these data augmentation techniques helped to increase the diversity of the dataset and improve the generalization ability of the model, resulting in more accurate and reliable classification of mpox images.

### 3.3. Model Selection and Training

Five pretrained deep learning models (VGG19, VGG16, ResNet50, MobileNetV2, and EfficientNetB3) were selected for assessment in this study. These models were trained using the preprocessed data, and their performance levels were evaluated using the accuracy, precision, recall, and F1-score metrics.

#### 3.3.1. Deep Convolutional Generative Adversarial Network (DCGAN)

The proposed technology, called DCGAN, was applied to generate synthetic data from real data. DCGAN is a widely used and efficient GAN network. In this research, we generated both real and synthetic data that are similar to the input data. The GAN architecture consists of two components, the generator and the discriminator, as shown in Figure 3. The generator creates synthetic samples, such as images and audio, with the goal of fooling the discriminator into mistaking them for real samples. In the first stage, the GAN network is prepared to create images from random noise inputs. Once a synthetic image is generated, both the synthetic and a real image are fed into the model for assessment. The distribution of our original and augmented data is shown in Table 2.

The work can be imagined by the drawing given below. In the discriminator model, images are used to classify whether the image is real or fake. Throughout the training period, the producer repeatedly tries to fool the discriminator by creating better fake images, whereas the discriminator is employed to discriminate between real and fake images. Numerous convolutional and convolutional–transpose layers are employed in both the discriminator and the generator. The steps included in the image augmentation algorithms for mpox detection are presented in Algorithm 1.
**Algorithm 1: Deep Learning Image Classification***//main steps in monkeypox image classification***Input**: Dataset**Output**: A network model: Model1.Begin
*//**Data Collection**: Data were gathered manually (45 images of monkeypox, 72 images of other diseases)*2.Data <-- **Normalization** (Data)3.Training, Validation, Test <-- **Split Data** (Data)
*//**Data processing**: Data augmentation (rotate, zoom in, zoom out, shift, share, and flip) using (TensorFlow. karas. preprocessing. Image)*
*Scale all images to (224 × 224)*4.For i = 1 to Number of Training do5.Data1 <-- **Horizontal Flip** (Training)6.Data2 <-- **Vertical Flip** (Training)7.Data3 <-- **ZoomIn** (Training)8.Data4 <-- **Zoom Out** (Training)9.Data5 <-- **Shift** (Training)10.//**Data Augmentation** <-- **Add** (Training, Data1, Data2, Data3, Data4, Data5)11.**generator** = make_generator_model ()12.**generated_image** = generator (noise, training)13.**discriminator** = make_discriminator_model ()14.**decision** = discriminator (generated_image)
//**Training model**: Divide the dataset into 80% for training and 20% for testing. Validation (chose 20% of images from training images)15Model <-- **Training Model** (Data Augmentation, Validation)16.Model <-- **Training Evaluation** (Model, Test)
***//Deep learning image classification (Determine if the photo contains monkeypox or not)****{EfficientNetB3, ResNet50, MobileNetV2, VGG16}*17.**End**

#### 3.3.2. Pretrained DL Models

The proposed method utilizes a deep learning approach to detect and classify mpox using image analysis. Five well-known pretrained CNN models, EfficientNetB3, ResNet50, MobileNetV2, VGG16, and VGG19, were used to improve the accuracy of mpox virus detection. We selected these models based on their outstanding classification performance, which was validated by computer vision. This includes their ability to accurately classify medical images when matched to our dataset.

I.VGG16 and VGG19

VGG-Net is a convolutional neural network that was developed by [19]. It consists of two models, VGG16 and VGG19, which performed well on the ImageNet dataset and placed first and second in the ImageNet2014 Visual Vision Challenge. These models have similar structures, with both consisting of convolutional filters (3 × 3), but they differ in terms of the number of layers. VGG16 has 16 layers and VGG19 has 19 layers. The model takes in an RGB image with dimensions of 224 × 224 and normalizes the pixel values from 0 to 255 to a range from 0 to 1. The image is then passed through a series of convolutional and fully connected layers, with VGG16 having 13 convolutional layers and 3 fully connected layers and VGG19 having 16 convolutional layers and 3 fully connected layers. The model also uses MaxPool 2D layers to reduce the image size and increase the depth of the filters. The final output of the model is a feature representation of the input image that can be used for classification or detection tasks.

An image size of 28 × 28 × 512 was used for both models. After that, the images were passed through 3 convolutional layers with 512 filters in VGG16, 4 convolutional layers with 512 filters in VGG19, and then a MaxPool 2D layer that reduced the image size to 14 × 14 × 512. Finally, the images were passed through 3 fully connected layers in both models with 4096 neurons in the first layer, 4096 neurons in the second layer, and 1000 neurons in the final layer. The final layer was used for classification, where the model output a probability for each class in the ImageNet dataset. In this work, we used VGG16 and VGG19 as pretrained models on the mpox image dataset and finetuned them to improve their performance in the mpox classification task.

II.ResNet

ResNet is a neural network that was first introduced by [20]. Its architecture was inspired by the architecture of the VGG-Net network. Multiple versions of ResNet have been developed, including ResNet50, which consists of 50 neural network layers. The main difference between ResNet50 and ResNet34 is that ResNet50 uses a stack of three layers instead of two. Figure 4 illustrates the network architectures for VGG16 and VGG19. ResNet was created to address the issue of performance degradation in models with a large number of layers. ResNet50 is particularly noteworthy for its 3.8 billion FLOPS, less than in VGG16 and VGG19, resulting in a lower cost. However, ResNet101 and ResNet152 have 101 and 152 layers, respectively, which increases their costs. To keep the performance cost reasonable, ResNet50 was adopted in this study and is preferred over ResNet101 and ResNet152 [18].

III.MobileNet

The MobileNet architecture was designed to reduce the computational cost of convolutional neural networks by using an inverted bottleneck instead of traditional 2D convolutional layers. Table 3 illustrates the structure of MobileNetV2, where the input images have dimensions of 224 × 224 × 3. The images pass through a 2D convolutional layer and then proceed through a series of bottleneck layers, resulting in dimensions of 7 × 7 × 320. They then pass through a final 2D convolutional layer and the avg. pool layer, resulting in dimensions of 1 × 1 × 1280 before being passed to the final step. SSD Light was introduced to reduce the cost of the process by reducing the number of parameters. Combining SSD Light with MobileNetV2 achieved competitive accuracy with fewer parameters.

IV.EfficientNetB3

EfficientNetB3 is a convolutional neural network [21] that balances the three components of a network: width, depth, and resolution. It is based on the concept of bottleneck convolution, similar to MobileNetV2, but with a larger FLOPS. The EfficientNet model suite was developed to explore the relationships among different basic network measurement dimensions under fixed resource constraints.

**Table 3 ijerph-20-04422-t003:** Network architectures for MobileNetV2 [22].

Input	Operator	T	c	n	s
224^2^ × 3	Conv2d	-	32	1	2
112^2^ × 32	Bottleneck	1	16	1	1
112^2^ × 16	Bottleneck	6	24	2	2
56^2^ × 24	Bottleneck	6	32	3	2
28^2^ × 32	Bottleneck	6	64	4	2
14^2^ × 64	Bottleneck	6	96	3	1
14^2^ × 96	Bottleneck	6	160	3	2
7^2^ × 160	Bottleneck	6	320	1	1
7^2^ × 320	Conv2d 1 × 1	-	1280	1	1
7^2^ × 1280	Avgpool 7 × 7	-	-	1	-
1 × 1 × 1280	Conv2d 1 × 1	-	k	-	

To optimize the efficiency of a model, it is often necessary to experiment with different options for the coefficients. We took several steps to ensure that our models could generalize well and avoid overfitting. To achieve this, we used a neural network architecture with 512 neurons and incorporated a dropout rate of 0.3. This enabled the model to handle inputs better and avoid over-reliance on specific classes or inputs. Hyperparameter tuning is a necessary process to optimize the efficiency of a model. It involves the adjustment of various parameters, such as the learning rate, regularization strength, and number of layers in the network, to determine which combination produces the best results. To further enhance the performance of our models, we started with a learning rate of 0.001 and created a subclass of Keras callbacks to regulate the learning rate. This allowed us to select the optimal learning rate based on the validation loss and improvements in the training accuracy to ensure that the models continually improved and adapted to the data. We also used a technique called finetuning, where the models were trained in two stages. We froze the top layers of the models in the first six epochs to allow the lower layers to develop better representations of the data. From epoch 7 onwards, we unfroze the top layers to enable the models to finetune the learned representations and further increase their performance. Once the hyperparameters were tuned, we evaluated the model’s performance on a validation set to prevent overfitting to the training data. If the performance was unsatisfactory, we adjusted the hyperparameters until the desired level of performance was achieved. Overall, these steps helped us to create models that could better generalize to new data and avoid overfitting, leading to a higher performance level in terms of the evaluation metrics. By taking a deliberate and thoughtful approach to model development, we were able to achieve results that outperform those of many other models in the field.

### 3.4. Evaluation and Comparison

The confusion matrix compares the predicted labels with the true labels and allows for the calculation of metrics such as the accuracy, precision, recall, and F1-score. The four elements included in the confusion matrix are true positives (TPs), true negatives (TNs), false positives (FPs), and false negatives (FNs). TPs represent correctly forecast positive cases, wherever the real class is mpox and the expected class is also mpox. TNs denote correctly predicted negative cases, where the actual class is not mpox and the predicted class is also not mpox. FPs represent instances where the actual class is not mpox but the predicted class is mpox. FNs represent instances where the actual class is mpox but the predicted class is not mpox. Table 4 shows the confusion matrix for the mpox classes. The model that performed the best was selected as the final model for mpox detection and classification. The performance levels of the four models were compared with the accuracy, precision, recall, and F1-score metrics.

Accuracy–accuracy is the proportion of correct predictions out of all predictions made by the model. Mathematically, it can be represented as
(1)$$\mathrm{Accuracy}=\frac{\mathrm{TP}+\mathrm{TN}}{\mathrm{TP}+\mathrm{FP}+\mathrm{FN}+\mathrm{TN}}$$

Precision–precision is the ratio of accurately forecast positive cases to all expected positive cases. In the context of this research, the precision is the ratio of correctly predicted mpox incidents to the total number of predicted mpox cases, including false positives. It can be represented mathematically as
(2)$$\mathrm{Precision}=\frac{\mathrm{TP}}{\mathrm{TP}+\mathrm{FP}}$$

Recall–recall is the ratio of correctly predicted positive cases to the total number of actual positive cases. In this research, recall represents the proportion of actual mpox cases that were correctly predicted by the model. Mathematically, it can be represented as
(3)$$\mathrm{Recall}=\frac{\mathrm{TP}}{\mathrm{TP}+\mathrm{FN}}$$

F1-Score—F1-Score is a weighted average of precision and recall that is considered to be a better metric than accuracy when the costs of false negatives and false positives differ. Mathematically, it can be shown as
(4)$$F1\;\mathrm{Score}=\frac{2\times \mathrm{Recall}\;\times \mathrm{Precision}\;}{\mathrm{Recall}\;+\;\mathrm{Precision}}$$

In situations where false negatives and false positives have the same cost, the F1-score is usually a better metric to use than accuracy. However, if their costs are different, it is better to consider both accuracy and recall.

## 4. Results

The classification performance of five pretrained deep learning models was evaluated, and the results are presented in Table 5. In this study, a dataset was used for the training, testing, and validation of the computer vision models. From the results in Table 5, it can be seen that the computation time for VGG16 was faster than that for ResNet50 while maintaining similar classification performance scores. Therefore, the best model weights with the lowest possible validation loss and the highest validation accuracy were loaded.

In this study, the performance of the model was evaluated with various metrics, including optimizer, loss function, and scoring metrics. The binary_crossentropy loss function was chosen since the classification was binary (i.e., monkeypox and non-monkeypox). The results showed that the VGG16 architecture performed the best among the models trained, achieving a training accuracy of 0.999 at epoch 9 and providing accuracy scores of 0.652, 0.984, 0.998, and 0.918 at epochs 13, 12, 14, and 14, respectively, for EfficientNetB3, VGG19, MobileNetV2, and ResNet50. Figure 5 illustrates the training and validation losses for the future deep neural network model. The plot shows the changes in the training accuracy and validation accuracy from epochs 7 to 14. The best epoch was chosen to avoid overfitting and underfitting. As shown in Figure 5, the training loss and validation loss for VGG16 decreased and stabilized at a specific point, indicating an optimal fit.

Figure 6 offers the confusion matrix for the different models used in the study, namely EfficientNetB3, ResNet50, VGG16, MobileNetV2, and VGG19. The matrix for each model is presented from (a) through (e). The confusion matrix represents the number of images that were categorized by the model as a particular class (actual), while they belonged to a various class (prediction).

### Comparison of the Performance of DL Models with Our Models

Table 6 illustrates the performance levels of the five models, where the best model was MobileNetV2, followed by VGG19, VGG16, ResNet50, and finally EfficientNetB3. The results show that the proposed model had the highest accuracy, precision, recall, and F1-score among all of the models tested. However, the MobileNetV2 model also performed marginally better than the suggested model, as seen by the accuracy (0.98), precision (0.99), recall (0.96), and F1-score (0.98). Figure 7 provides a graphical representation of the performance levels of the classifier models and compares them.

## 5. Discussion

The models were evaluated using our own dataset, which showed promising results in terms of the accuracy, precision, recall, and F1-score when compared to most other research conducted in the field. We further evaluated the models using the MSLD dataset, which was collected by another research group, and the results showed that our models performed better than theirs, thus confirming the effectiveness of our models. In particular, we compared our proposed model to other pretrained DL models, including EfficientNetB3, VGG19, VGG16, ResNet50, and MobileNetV2. The evaluation results revealed that our proposed model outperformed all other models in terms of its accuracy, precision, recall, and F1-score. However, the MobileNetV2 model achieved slightly better results in terms of the evaluation metrics. To further validate the performance of our proposed model, we also tested it on the MSLD dataset [5]. The results showed that our proposed model and other models performed well on this dataset, with MobileNetV2 achieving the highest accuracy score of 0.9890. Therefore, the evaluation results demonstrate that the proposed model is competitive and has a good fitness level when used with the given dataset when compared with other deep learning models.

To the best of our knowledge, there has been limited research on the use of deep learning for the detection of monkeypox. One notable exception is the work of Adekunle et al. (2020), who proposed a deep learning approach for the detection of monkeypox from skin lesion images. However, their approach was limited to a small dataset and did not make use of transfer learning. Our models were trained using both complete augmented data and augmented training data. This approach allows for better performance when there are fewer instances available. While augmentation was applied to the entire dataset, the validation and testing data became imbalanced, with most cases being non-mpox. On the other hand, when augmentation was just applied to the training data, the validation and testing data also became imbalanced, but most cases were mpox. Despite the imbalance in the data, the models performed well with only one case being misclassified. As seen in Table 1, the suggested model accomplished the highest performance level.

### 5.1. Validation of Our Model with Another Dataset

This section presents the validation of 20% of the dataset using another dataset [5]. Table 7 compares the performance levels of five methods that focus on mpox detection using different image datasets and are evaluated using various metrics. The results of these models for the current dataset show a high level of performance, particularly the MobileNetV2 model, followed by VGG19. As seen in Table 7 and Figure 8, the MobileNetV2 model achieved the highest performance level with an accuracy score of 0.94.

### 5.2. Comparison with Other Studies

This part presents an assessment of the DL model for detecting mpox positive cases and compares the results with related research [6,7,8,9,10,11]. The models used in this comparison are based on transfer learning and are standardized in terms of conditions and parameters. The techniques identified as the best in the challenge were chosen and associated with the recommended method. Previous studies focused on mpox detection using different image datasets and evaluation metrics. A comparison of similar DL approaches with the proposed model was conducted. Most of the datasets had a limited number of images for training and testing, which made it difficult to create and improve their models. The most common methods used for model creation in these studies were based on VGG and ResNet. In this study, we employed VGG16, ResNet50, MobileNetV2, and InceptionResNetV2 to make the model faster and more reliable, making it suitable for use as a real-time assessment tool. Ultimately, linked to other techniques, the suggested method with the MobileNetV2 model offers higher levels of accuracy (99%) and efficiency.

Our model can successfully detect both mpox and non-mpox. Our algorithm can detect the disease with high accuracy and has the potential to be a valuable tool for quick and accurate diagnosis of mpox in clinical settings. The suggested technique could be a valuable tool for rapidly and accurately diagnosing mpox in clinical settings. The main advantage of the proposed model is that it achieved a high level of accuracy when detecting monkeypox cases, 99%. The model uses the MobileNetV2 architecture, which is a quick and efficient model for image classification tasks. The suggested method was found to perform better than those described in previous studies, and the authors believe that this model could be used for the real-time detection and forecasting of monkeypox cases. We hope that this work will aid in the quick discovery, classification, and management of monkeypox, making it a valuable tool for quick and accurate diagnosis in clinical settings.

The main contribution of our work is that we described a state-of-the-art machine learning technique for the early detection of monkeypox, which can significantly improve the speed and accuracy of diagnosis. This could ultimately aid in the quick discovery, classification, and management of monkeypox. The high level of accuracy achieved by the proposed model in classifying monkeypox in both the training and test sets shows that it is a potentially valuable tool for quick and accurate diagnosis in clinical settings. The results of this study could enable medical professionals to diagnose a wide range of diseases quickly and accurately, leading to improved patient outcomes and reduced healthcare costs. Additionally, the use of the MobileNetV2 model represents a departure from previous studies that primarily used the VGG and ResNet models. The proposed model demonstrates better accuracy and efficiency levels than these models. The proposed model also has the potential to be used as a real-time assessment tool, and it could be implemented on smartphones for the real-time detection and forecasting of monkeypox cases. The high level of accuracy achieved by the proposed model in the classification of monkeypox in both the training and test sets makes it a potentially valuable tool for quick and accurate diagnosis in clinical settings. Furthermore, the research on automated monkeypox detection using deep learning and transfer learning techniques has the potential to make significant contributions to the field of healthcare and disease diagnosis and could potentially lead to the development of new diagnostic tools and techniques for other infectious diseases.

The proposed work has some limitations that should be considered when evaluating its results. One limitation is that the model was trained and tested on a limited dataset, which may not be representative of all cases of mpox. Additionally, all data were collected internally, and it would be beneficial to validate the model on other external datasets to confirm its accuracy. Further evaluation and validation of the model with larger and more diverse datasets in real-world scenarios would be valuable to determine its capabilities and limitations. Such evaluations could assist in determining whether the model is suitable for use as a real-time diagnostic tool for monkeypox in clinical settings.

The use of deep learning models for the early detection and diagnosis of monkeypox could provide several benefits for managers. It could lead to prompt medical intervention and improved patient outcomes, resulting in reduced healthcare costs. Professionals in the medical field could benefit from the use of this model by being able to diagnose a wide range of diseases quickly and accurately. Additionally, deep learning models can be trained on large datasets to improve their accuracy and efficiency when detecting and classifying monkeypox, leading to more reliable and efficient diagnosis of the disease. The use of deep learning models for monkeypox detection and classification could lead to cost savings by reducing the need for more expensive and time-consuming diagnostic methods. The use of the MobileNetV2 model also represents a departure from previous studies and was demonstrated to have better accuracy and efficiency levels than other models. Moreover, the proposed model has the potential to be used as a real-time assessment tool that can be implemented on smartphones for the real-time detection and forecasting of monkeypox. This can greatly aid in the quick discovery, classification, and management of monkeypox for both managers and patients, ultimately leading to improved health outcomes.

## 6. Conclusions

The objective of this research was to improve DL-based methods for mpox detection with the goal for achieving a higher accuracy than that of existing algorithms. A novel method using the MobileNetV2 model was proposed and found to have the highest performance level with an accuracy score of 99%. The suggested method was found to perform better than models described in previous studies. The potential future application of this model is the real-time detection and forecasting of mpox on smartphones. To further prove the feasibility of the model, it may be necessary to study larger image sizes in future studies. We hope that this work and future related research will aid in the quick discovery, classification, and management of mpox. Our results are promising, suggesting that our model could be a state-of-the-art machine learning technique for the early detection of mpox. Our algorithm was able to achieve a high level of accuracy when classifying mpox in both the training and test sets, making it a potentially valuable tool for quick and accurate diagnosis in clinical settings.

## Figures and Tables

**Figure 1 ijerph-20-04422-f001:**
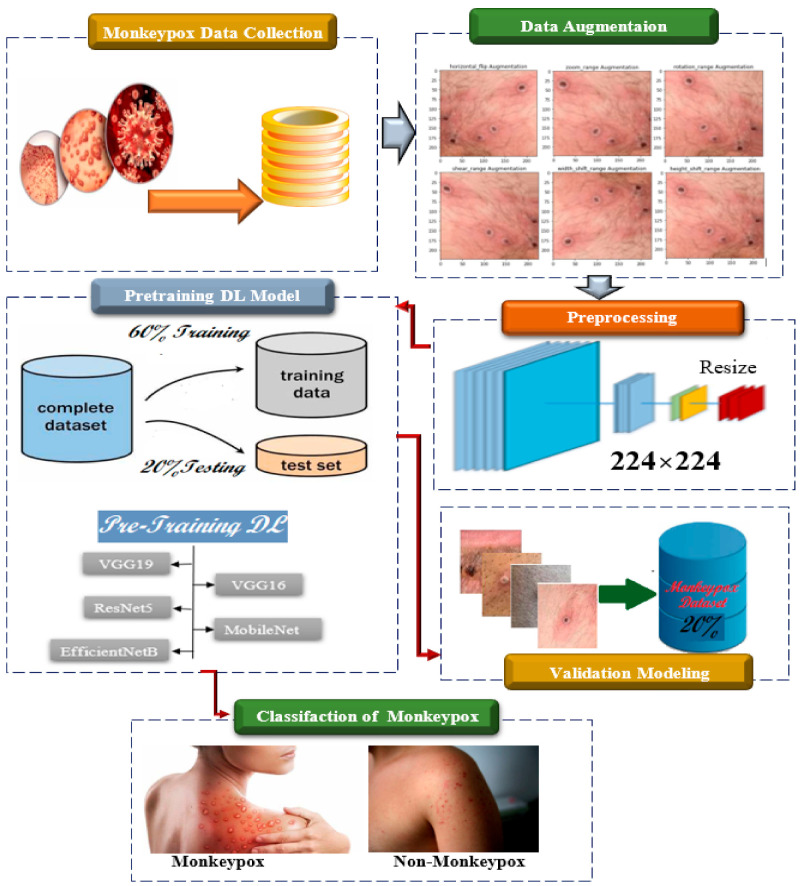
The steps and procedures included in the proposed approach.

**Figure 2 ijerph-20-04422-f002:**
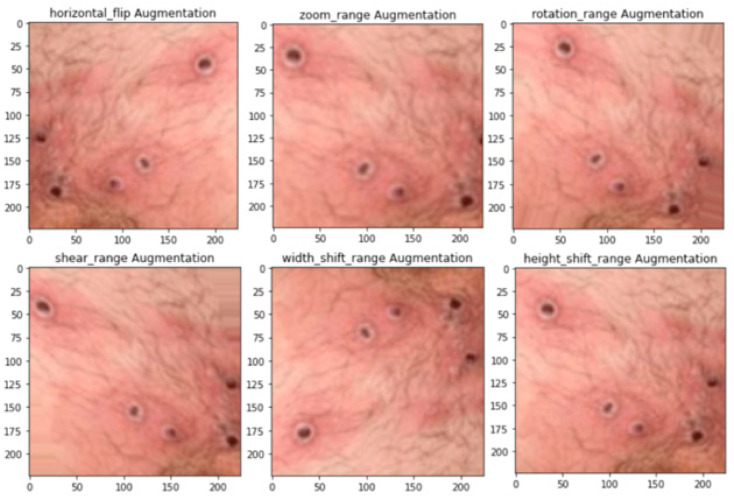
Data augmentation methods.

**Figure 3 ijerph-20-04422-f003:**
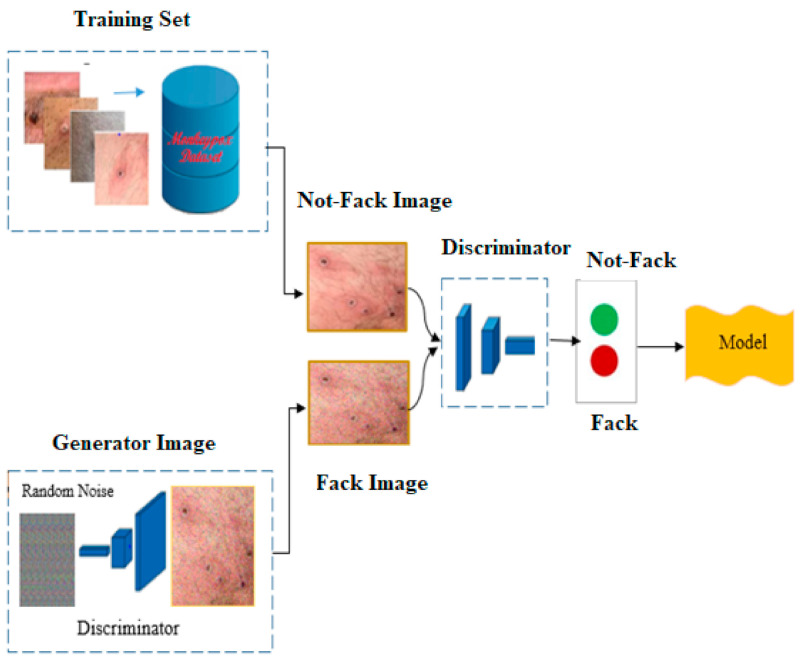
AGN network model.

**Figure 4 ijerph-20-04422-f004:**
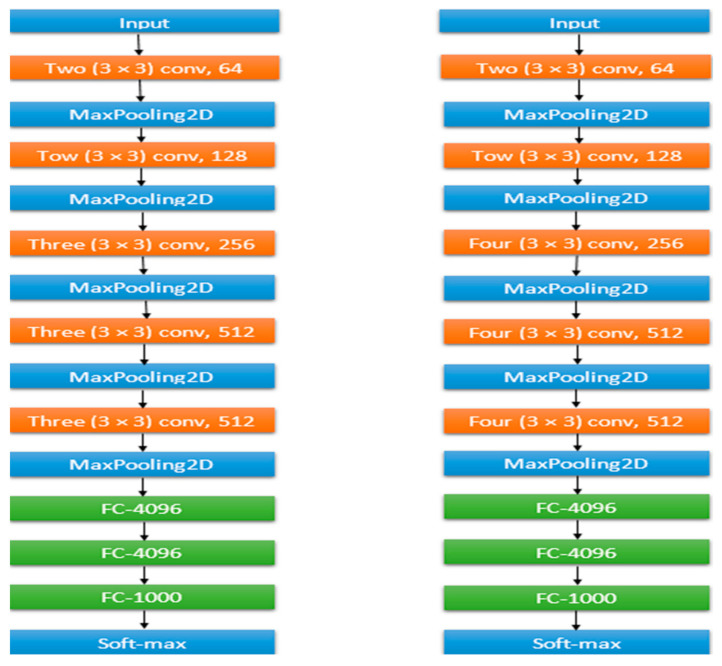
Network architectures for VGG16 and VGG19.

**Figure 5 ijerph-20-04422-f005:**
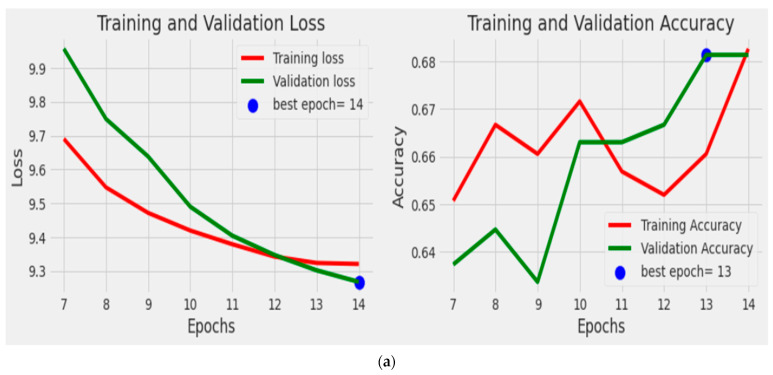
Training and validation accuracy and loss for our DNN method: (**a**) EfficientNetB3; (**b**) ResNet50; (**c**) MobileNetV2; (**d**) VGG16; (**e**) VGG19.

**Figure 6 ijerph-20-04422-f006:**
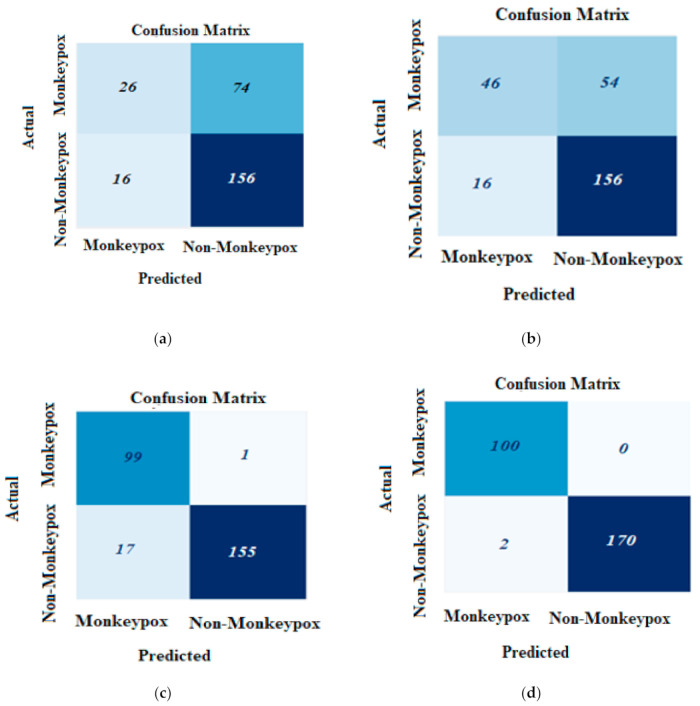
Confusion matrix for our model: (**a**) EfficientNetB3; (**b**) ResNet50; (**c**) VGG16; (**d**) MobileNetV2; (**e**) VGG19.

**Figure 7 ijerph-20-04422-f007:**
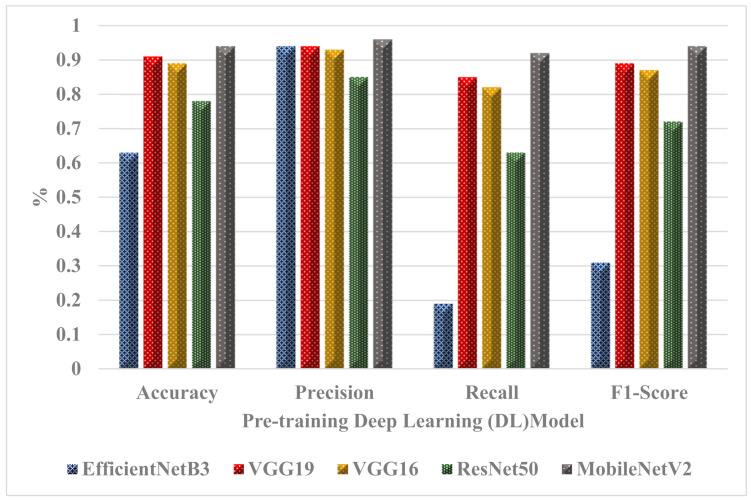
Performance of the classifier models.

**Figure 8 ijerph-20-04422-f008:**
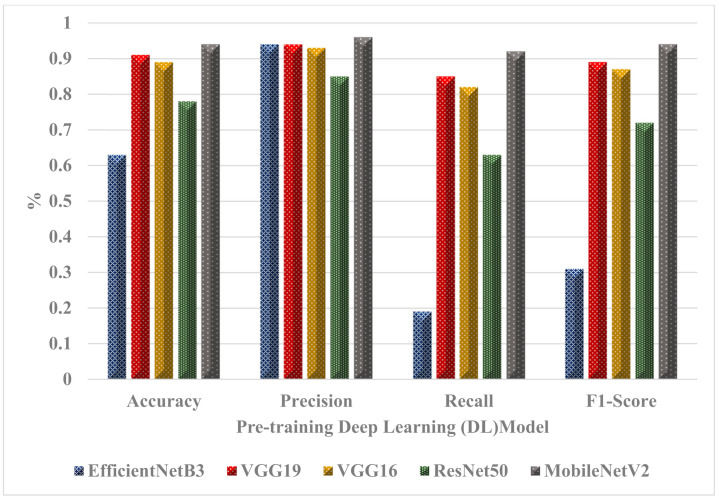
Performance of the classifier models.

**Table 1 ijerph-20-04422-t001:** Summary of work related to mpox detection.

Author	Approaches	Best Results	Validation	Limitation and Gaps
[7]	VGG-16, ResNet50, and InceptionV3	Accuracy: 82.96%	70% training 10% validation 20% testing	Low accuracy and it can improve
[8]	ResNet18, GoogleNet, EfficientNetB0, NasnetMobile, ShufeNet, and MobileNetV2	Accuracy: 91.11%.	70% training, 20% testing. 10% validation	Low accuracy and small dataset
[9]	VGG16	AUC: 97.2%	80% training 20% testing	They did not use different models
[10]	VGG-16, VGG-19, ResNet50, ResNet101, IncepResNetv2, MobileNetV2, InceptionV3, Xception, EfficientNetB0, EfficientNetB1, EfficientNetB2, DenseNet-121, and DenseNet-169	Avg. Accuracy: 85.44%	No Validation	Low accuracy
[11]	CNN models AlexNet, GoogleNet and Vgg16Net with, Naïve Bayes, SVM, KNN, Random Forest (RF), and Decision Tree (DT)	Accuracy: 91.11%	No Validation	No graph to confirm the results
[12]	Polynomial Regression, SVR, Holt’s Linear Model AR Model, SARIMA Model ARIMA Model, MA Model, Holt–Winter’s Model, and Prophet Model	MSE: 41,922.55 R2: 0.49 MAPE: 16.82 MAE: 146.29 RMSE: 204.75	No Validation	Low accuracy
[13]	Binary PSOBER algorithm	Accuracy: 98.8%	No Validation	No limitation
[14]	CNN model based on MobileNetV3-s, EfficientNetV2, ResNet50, Vgg19, DenseNet121, and Xception models.	Accuracy: 96%.	No Validation	No limitation
[15]	CNN and LSTM	Accuracy: 94%.	70% training, 30% validation	No limitation

**Table 2 ijerph-20-04422-t002:** Distribution of our original and augmented data.

Type of Image	Original Images	Augmented Images
Monkeypox	45	540
Normal	25	300
Scarlet fever	22	264
Roseola	25	300

**Table 4 ijerph-20-04422-t004:** Confusion matrix of the mpox classes.

		Prediction	
		Positive: Monkeypox	Negative: No-Monkeypox
Actual	Positive: Monkeypox	Number of TP	Number of FN
	Negative: No-Monkeypox	Number of FP	Number of TN

**Table 5 ijerph-20-04422-t005:** Training and validation times with the best accuracy levels.

Classifier	Training Time (Second)	Testing Time (Second)	Epoch Number	Best Epoch	Per Epoch	Best Training Accuracy	Best Validation Accuracy
EfficientNetB3	264	5	14	13	11	0.652	0.667
VGG19	297	9	14	12	11	0.984	0.934
VGG16	304	9	14	11	11	0.999	0.981
ResNet50	269	5	14	14	11	0.918	0.733
MobileNetV2	256	4	14	9	11	0.998	0.992

**Table 6 ijerph-20-04422-t006:** Performance of the classifier models.

Classifier Model	Accuracy	Precision	Recall	F1-Score
EfficientNetB3	0.6838	0.38	0.72	0.49
VGG19	0.9779	0.96	0.98	0.97
VGG16	0.9669	0.95	0.96	0.95
ResNet50	0.7069	0.34	0.90	0.49
MobileNetV2	0.9816	0.99	0.96	0.98

**Table 7 ijerph-20-04422-t007:** Performance of the model validation with the MSLD dataset [5].

Classifier of Models	Accuracy	Precision	Recall	F1-Score
EfficientNetB3	0.63	0.94	0.19	0.31
VGG19	0.91	0.94	0.85	0.89
VGG16	0.89	0.93	0.82	0.87
ResNet50	0.78	0.85	0.63	0.72
MobileNetV2	0.94	0.96	0.92	0.94

## Data Availability

Data sharing does not apply to this article, as no datasets were generated during the current study.

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
