# Peer review of "Automated Monkeypox Skin Lesion Detection Using Deep Learning and Transfer Learning Techniques"

_ijerph, 2023, doi:10.3390/ijerph20054422_

Round 1
Reviewer 1 Report
Jaradat et al. estimate the performance of different deep-learning object detection models for mpox images. They successfully finished data collection, data augmentation, model training, validation, and testing for all 5 models which provides the foundations for the community to access the potential of deep-learning-based models for mpox detection.
But I have the following major concerns,
-
The author claims their work to be an algorithm, e.g., “Our algorithm” in line 39 and line 533, however, based on my understanding the novelty is that they have summarized a complete how-to step guide as they summarized in line 361 Algorithm 2. They need to justify what is new in their algorithm.
-
Also, in line 280 Algorithm 1 was called Image Augmentation, but it contains more than image augmentation, for example, training models and evaluating models are also included. Another better name shall be given to this part also this is a how-to step guide, not an algorithm and these are all common practices in object detection.
-
All those investigated 5 models are relatively old, VGG16 and VGG19 are 2015, REsNet50 is 2016, MobileNetV2 is 2018, and EfficientNet is 2019. How about the latest models developed since 2020, for example, Transformer based models like DETR(End-to-End Object Detection with Transformers) or YOLO, Faster R-CNN, etc? Even if we ignore other types of models, how about U-Net-based models? In medical applications, I would say the U-Net family is the most popular choice and can be translated to object detection too.
-
In Line 430, the authors present the training curves for EfficientNetB3 and claim that Epoch 14 is the best. However, Epoch 14 enters the overfitting stage, based on the figure, Epoch 12 is better, since it is the cross-over point of training loss and validation loss. They might also explain why EfficientNetB3 is always the worst model.
I also have the following minor suggestions for the author, generally please do a thoroughly proofread since there are a lot of grammar and typos.
-
It would be better to replace all “monkeypox” with “mpox” to follow the suggestion from WHO, refer to this WHO webpage for more information, https://www.who.int/news/item/28-11-2022-who-recommends-new-name-for-monkeypox-disease
-
Line 51, “transmitted to humans from animals in the Democratic in 1970”, here I guess the author wants to say Democratic Republic of the Congo (République démocratique du Congo), not “Democratic”. https://en.wikipedia.org/wiki/Mpox
-
Lines 61 to 63 is exactly the same meaning as lines 65 to 67 where 61-63 is “However, early detection is difficult due to the similarity between monkeypox and other diseases such as scarlet fever, Roseola, and smallpox.” and 65-67 is “However, early detection of monkeypox is difficult due to its similarity to other diseases such as scarlet fever, Roseola, and smallpox.” Remove duplicates.
-
Line 74, DL shall present full word format before using an acronym.
-
In line 87, the author uses “Monkey-pox” but in most texts, they use “monkeypox”. Please use only 1 format for one word of the same meaning.
-
In most cases, there is no empty line between paragraphs in the same section, why is there an empty line in Line 152?
-
Line 262, the text is like “into for evaluation.Distribution of our original and augmentation data as shown in”, besides the typo “showen”, I noticed there is no space between two sentences, which happens a lot of times in the article. For example, line 341, “resulting in dimensions of (1 x 1 x 1280) before being passed to the final step.SSD”.Please carefully proofread to correct them.
-
Line 306, Figure 4 has the wrong caption. In the text, the author says “Figure 4 illustrates these differences. ResNet was created to address the issue of performance degradation in models with a large number of layers.”, however in Figure 4 they say in the caption, “Network architectures for VGG16 and VGG19.”.
-
Line 458, “Table 6 illustrates the performance of the five models” however, in most places, the author will say, “Table.6”. Actually, both formats are fine, but we shall not use two formats in the same article.
Author Response
kindly see the attached response

Reviewer 2 Report
1. What is the main advantage of the proposed model?
2.What is the main contribution to the field from your side? Make a better presentation of standard ideas and late advances in the field.
3. The introduction and related work section could be extended and incorporate additional discussions on monkeypox detection, deep learning, and transfer learning approaches, e.g., https://www.mdpi.com/1424-8220/23/4/1783
https://arxiv.org/abs/2208.12019
https://www.ncbi.nlm.nih.gov/pmc/articles/PMC9548428
https://arxiv.org/pdf/2207.03342. This could set the scene and background for the subsequent discussions in this manuscript.
4. What are the limitations of the proposed work? Can you show the operation of your model in various scenarios?
5. Your paper does not show details of the model composition. How do you evaluate the fitness of the proposed model?
6. How to set optimal values of coefficients in your model? Did you test different options for coefficients to optimize the efficiency of the model, etc.?
7. What are the managerial benefits of the analyses?
Author Response
kindly see the attached response

Round 2
Reviewer 1 Report
I have no questions and the author addressed my concerns.
Reviewer 2 Report
Author has done required changes. Paper may be accepted.